# Identification and Functional Analysis of Apoptotic Protease Activating Factor-1 (Apaf-1) from *Spodoptera litura*

**DOI:** 10.3390/insects12010064

**Published:** 2021-01-13

**Authors:** Haihao Ma, Xiumei Yan, Lin Yan, Jingyan Zhao, Jiping Song, Rong Peng, Yongbo Yang, Jianxin Peng, Kaiyu Liu

**Affiliations:** School of Life Sciences, Central China Normal University, Wuhan 430079, China; mhh8486@mails.ccnu.edu.cn (H.M.); yanxiumei77@163.com (X.Y.); ylmorning@139.com (L.Y.); jingyan_zhao@163.com (J.Z.); songjiping2007@163.com (J.S.); pengrong@mail.ccnu.edu.cn (R.P.); yongboyang@mail.ccnu.edu.cn (Y.Y.)

**Keywords:** Apaf-1, apoptosis, caspase, *Spodoptera litura*

## Abstract

**Simple Summary:**

Apoptosis plays an important role in both the development of lepidopteran insects and the elimination of cells. The apoptosis signal pathways are well documented in mammals and *Drosophila melanogaster*. However, it remains less clear in lepidopteran insects. This study characterized the apoptotic protease activating factor-1 (Apaf-1) from *Spodoptera litura*. The results showed that *S. litura* Apaf-1 (Sl-Apaf-1) is similar to the mammalian Apaf-1. Sl-Apaf-1 consists of a caspase recruitment domain (CARD), as well as nucleotide-binding and oligomerization domain (NOD) and the C-terminal WD40-repeat domain (WD), and interacts with Sl-caspase-5 (a homologue of mammalian caspase-9). The activated Sl-caspase-5 can cleave Sl-procaspase-1 (a homologue of caspase-3 in mammals), which directly causes apoptosis. The apoptosis signal pathway is conserved between lepidopteran insects and mammals.

**Abstract:**

Apoptotic protease activating factor-1 (Apaf-1) is an adaptor molecule, essential for activating initiator caspase and downstream effector caspases, which directly cause apoptosis. In fruit flies, nematodes, and mammals, Apaf-1 has been extensively studied. However, the structure and function of Apaf-1 in Lepidoptera remain unclear. This study identified a novel Apaf-1 from *Spodoptera litura*, named Sl-Apaf-1. Sl-Apaf-1 contains three domains: a CARD domain, as well as NOD and WD motifs, and is very similar to mammalian Apaf-1. Interference of *Sl-apaf-1* expression in SL-1 cells blocked apoptosis induced by actinomycin D. Overexpression of *Sl-apaf-1* significantly enhances apoptosis induced by actinomycin D in Sf9/SL-1/U2OS cells, suggesting that the function of Sl-Apaf-1 is evolutionarily conserved. Furthermore, Sl-Apaf-1 could interact with Sl-caspase-5 (a homologue of mammalian caspase-9) and yielded a binding affinity of 1.37 × 10^6^ M^–1^ according isothermal titration calorimetry assay. Initiator caspase (procaspase-5) of *S. litura* could be activated by Sl-Apaf-1 (without WD motif) in vitro, and the activated Sl-caspase-5 could cleave Sl-procaspase-1 (a homologue of caspase-3 in mammals), which directly caused apoptosis. This study demonstrates the key role of Sl-Apaf-1 in the apoptosis pathway, suggesting that the apoptosis pathway in Lepidopteran insects and mammals is conserved.

## 1. Introduction

Apoptosis is a highly conserved mechanism by which eukaryotic cells commit suicide. This mechanism plays a critical role in tissue development, tissue homeostasis, host defense, and the elimination of unwanted cells in multicellular organisms [1,2]. Apoptosis is also associated with many human diseases [3]. In mammals, two major apoptotic signaling pathways, intrinsic and extrinsic, have been well characterized [4,5,6]. The intrinsic pathway is also known as the mitochondrial pathway. Cytochrome c, apoptotic protease activating factor-1 (Apaf-1), and dATP form the apoptosome, which can recruit and activate caspase-9, which in turn triggers apoptosis by initiating the caspase-3 dependent proteolytic cascade [7]. The extrinsic pathway is mediated by death receptor. If activated, the death receptor can recruit adaptor proteins and caspase-8/10. This pathway triggers apoptosis through cleavage of downstream effector proteins, such as caspase-3/6/7 [8]. Apaf-1 was first identified in HeLa cells and was found to be a homologue of *Caenorbabditis elegans* CED4 [9]. It contains a caspase recruitment domain (CARD) at the N terminus, a nucleotide-binding and oligomerization domain (NOD), and two WD40 repeat domains at the C terminus. The CARD is the binding module for procaspase-9, the NOD hosts a nucleotide binding site and mediates the oligomerization into the apoptosome, and the regulatory WD40-repeat domain serves as binding site for cytochrome c [10]. Both the activation and acting mode of Apaf-1 in the mammalian apoptosis signal pathway are well understood [11]. Briefly, Apaf-1 binds to cytochrome c, released from mitochondria, and dATP to form oligomeric apoptosome, which may induce caspase-9 dimerization and subsequent autocatalysis. Activated caspase-9 stimulates the subsequent caspase cascade, thus committing the cell to apoptosis. Biochemical tests showed that the expression of the CARD dimer of Apaf-1 could bind and activate the caspase-9 protein [12], and the expression of Apaf-1 without WD40 motif could lead to apoptosis independent of the presence of cytochrome c [13]. The Apaf-1 gene in *Plutella xylostella* and *Bombyx mori* have been cloned [14,15], but the structure and functions of Apaf-1 in Lepidopteran have not been investigated to date.

Apoptosis also plays an important role in insect physiology, e.g., development, tissue homeostasis, DNA damage response, and immune response [16]. Evidence suggests that the mitochondrial apoptosis pathway also exists in lepidopteran insects (e.g., *Spodoptera frugiperda*, *Spodoptera exigua*, *Spodoptera litura*, and *Bombyx mori*), and studies from different laboratories have shown that cytochrome c plays an essential role during apoptosis [15,17,18,19,20,21,22,23,24,25,26]. Furthermore, two important caspases in the mitochondrial apoptosis pathway were identified in Lepidoptera. Caspase-1 is the most studied effector caspase, and its activation may lead cells to apoptosis under UV irradiation or as a result of baculovirus infection. In Sf9 cells, caspase-1 is processed during apoptosis to mature subunit fragments in a two-step activation mechanism: the first step is the cleavage of proenzyme (P37) at D195 (which produces P25 and the small subunit P12) while the second stage is the cleavage of P25 at D28 (which producing the prodomain P6 and the large subunit P19). The same cleavage was found in SL2 cells (from *Spodoptera littoralis*) for Sl-caspase-1 [27,28]. Caspase-5 (Dronc) was identified as an initiator caspase in *S. frugiperda*, and recombinant Sf-caspase-5 can cause autocatalytic cleavage at position D340 and cleaves the Sf-caspase-1 [29]. However, it remains unclear how the apoptotic machinery responds to the mitochondrial apoptosis signal and activation of caspase because the function of Apaf-1 has not been identified to date. Therefore, the identification and functional analysis of Apaf-1 are important in the apoptosis of lepidopteran insects.

This study identified an *apaf-1* homolog gene from *S. litura* and designated it *Sl-apaf-1*. The function of Sl-Apaf-1 during apoptosis was studied through RNA interference and over-expression assay. Protein expression and biochemical analysis were also performed to further characterize Sl-Apaf-1. The results indicate that Sl-Apaf-1 plays an important role in apoptosis, and provide useful information for the study and utilization of the apoptosis pathway in Lepidoptera.

## 2. Materials and Methods

### 2.1. Cell Culture and Antibodies

Cell line SL-1, which originated from an embryo of *S. litura* (SL-ZSU,) was obtained from the Institute of Entomology, Sun Yat-sen University, Guangzhou, Guangdong, China. Sf9 (*S. frugiperda* pupal ovarian tissue cells) and U2OS (*Homo sapiens* bone osteosarcoma cells) cells were maintained in the laboratory. The insect cell lines were cultivated in Grace’s Insect Medium (Gibco, Carlsbad, CA, USA) supplemented with 8% fetal bovine serum (Gibco), 100 U/mL penicillin and 100 µg/mL streptomycin (Gibco) at 28 °C. U2OS cells were grown in Dulbecco’s Modified Eagle Medium (DMEM, Gibco) supplemented with 10% fetal bovine serum, 100 U/mL penicillin, and 100 µg/mL streptomycin (Gibco) at 37 °C and 5% CO_2_. β-Actin Rabbit mAb (#12620) and Anti-rabbit IgG, HRP-linked antibody (#7074) were purchased from Cell Signaling Technology (CST, Danvers, MA, USA). Anti-Sf-caspase-1 antiserum was provided by Paul Friesen [28].

### 2.2. Cloning of Sl-apaf-1 Gene

Sequence alignment of putative *Apaf-1* from *B. mori, Manduca sexta,* and *Danaus plexippus* was performed by Clustal W software version 2.0 (https://www.ebi.ac.uk/Tools/msa/clustalw2/). Highly conserved nucleotide sequences were used to design and synthesize specific primers (Figure 1). A 778 bp fragment of *Sl-apaf-1* was first obtained by RT-PCR using primers F1 and R1. Based on this piratical sequence, a 653 bp fragment upstream and a 3057 bp fragment downstream were amplified by semi-nested PCR using specific primer pairs (F2/R2-1 and F2/R2-2 for the upstream sequence; F3-1/R3, F3-2/R3, and F3-3/R3 for the downstream sequence). The remaining cDNA sequences of *Sl-apaf-1* were amplified by 3′- and 5′-RACE using the Smart RACE cDNA amplification kits (Clontech, Palo Alto, CA, USA). The experiment was performed according to the recommendation of the manufacturer, and the gene specific primers (GSP) are listed in Table 1. The full-length coding sequence (CDS) of S*l-apaf-1* was amplified with Phusion^®^ High-Fidelity DNA Polymerase (Thermo Scientific, Waltham, MA, USA). PCR was performed under the following conditions: pre-denaturation at 98 °C for 30 s; then, 30 cycles of heating to 98 °C for 10 s, cooling to 60 °C for 15 s, and re-heating to 72 °C for 120 s; followed by a final 5 min cycle at 72 °C. The PCR product was separated using gel electrophoresis (0.7% agarose), and then recovered and cloned into the pGEM-T vector (Promega, Madison, WI, USA) by TA clones. The recombinant plasmid (pGEM-Sl-Apaf-1) was then transformed into competent *Escherichia coli* DH5α cells and positive clones were screened on an LB agar plate (with 100 μg/mL ampicillin) for plasmid extraction and sequence verification.

The restriction endonucleases Xho I and Pst I were used to digest and isolate the CDS of *Sl-apaf-1* from the pGEM-Sl-Apaf-1 vector. The ligation product (pEGFP-N1–Sl-Apaf-1) was constructed by subcloning the digested *Sl-apaf-1* CDS into a pEFGP-N1 vector. To express Sl-Apaf-1 in insect cells, the pIE2-Sl-Apaf-1-EGFP vector was constructed by replacing the CMV promoter in the pEGFP-N1–Sl-Apaf-1 vector with the OpIE2 promoter from the pIZTV5-His vector.

### 2.3. Structure and Phylogenetic Analysis of the Sl-Apaf-1 Protein

The I-Tasser server (https://zhanglab.ccmb.med.umich.edu/I-TASSER/) was used to predict the tertiary structures of the putative Sl-Apaf-1 protein [30]. PyMOL software version 1.8.0.0. (http://www.pymol.org/) was used to visualize the protein structure, and the subdomain schema was made by DOG software (Domain Graph, version 1.0, http://dog.biocuckoo.org/). For phylogenetic analysis, 10 homologous Apaf-1 proteins sequences were download from GenBank. MEGA version 7.0.21 was used to construct the tree (bootstrap test: 1000 replications; model: Amino: Poisson correction; gaps: treated by pairwise deletion) [31].

### 2.4. Cell Transfection and RNA Interference of SL-1 Cells

The SL-1 cells were seeded into 6-well plates at 5 × 10^5^ cells/well and were cultured overnight. Cells in each well were transfected with 2 μg of plasmid and 8 μL of cellfectin reagent II (Invitrogen, Palo Alto, CA, USA) according to the manufacturer’s protocol. The cells were continuously cultured for 48 h for detection. For RNA interference, siRNA sequences that target S*l-apaf-1* mRNA were designed (Table 2) and were synthesized by GenePharma (GenePharma Co., Ltd., Shanghai, China). These siRNAs (40 pmol per well) were transfected into SL-1 cells using cellfectin II (Invitrogen) and total RNA was extracted using the TRIzol Reagent (Applied Invitrogen, Carlsbad, CA, USA). One microliter of total RNA sample was used for the synthesis of first strand cDNA with PrimeScript RT regent kit (Takara, Shanghai, China). A 15-ng cDNA sample, which was reversely transcribed from total RNA, was used as template for qRT-PCR with SYBR Premix Ex Taq II Kit (Takara). S*l-apaf-1* and glyceraldehyde-3-phosphate dehydrogenase (*gapdh*, HQ012003) were amplified using specific primers (see Table 1). qRT-PCR was performed on a CFX96 Touch Real-Time PCR Detection System (Bio-Rad, Hercules, CA, USA). The qRT-PCR conditions were as follows: pre-denaturation at 95 °C for 1 min, followed by 95 °C, for 10 s, and 60 °C for 30 s and 40 cycles. Fluorescent signals from each sample were collected at the end of each cycle. The expression level of *Sl-apaf-1* was normalized according to the internal *GAPDH* control [32], and results were plotted as relative values. The fold changes between the treated groups and a normal control were calculated using the 2^−ΔΔCT^ method [32]. qRT-PCR was repeated in triplicate for each sample.

### 2.5. Western Blot Analysis

Actinomycin D (Act D) is a well-known clinical antitumor drug that can induce the apoptosis of HepG2 through the Fas- and mitochondria-mediated apoptosis pathway [33]. It can also induce extensive and rapid apoptosis in the cells of lepidopteran insects [34]. SL-1 cells were treated with 500 ng/mL Act D (Sigma, Saint Louis, MO, USA) for 6 h. The total protein of cells was extracted with RIPA Lysis Buffer (Beyotime, Shanghai, China) and was quantified with BCA protein assay (Pierce Biotechnology, Rockford, IL, USA). A total of 40 μg of each sample was separated by 10% SDS-PAGE and was transferred to 0.45 μm Immobilon-P PVDF membranes (Millipore Corporation, Billerica, MA, USA). The membranes were blocked with TBST buffer (25 mM Tris, 150 mM NaCl, 0.02% Tween-20, at a pH range of 7.2–7.5) and 5% non-fat milk for 40 min. After washing three times with TBST buffer, the membranes were incubated with primary antibodies against Sl-caspase-1 and actin (Cell Signaling Technology, Danvers, MA, USA) at a 1:2000 dilution in 1% BSA for 2–3 h. After washing with TBST, the membranes were further incubated at room temperature for 1 h with horseradish peroxidase-conjugated secondary antibodies (Cell Signaling Technology) in blocking buffer and were then stained with ECL Plus (Beyotime).

### 2.6. Microscopy

The transfected cells were digested and replanted in 6-well plates with cover glass. After the cells had adhered to the cover glass, they were washed with PBS buffer once (0.24 g NaH_2_PO_4_, 1.44 g Na_2_HPO_4_, 8 g NaCl, and 0.2 g KCl in 1 L water, at pH 7.4), and fixed with 40% paraformaldehyde for 10 min. Then, cells were washed with PBS once, and stained with 0.1 μg/mL 4′,6-diamidino-2-phenylindole (DAPI) in PBS for 10 min. After staining, the cells were washed three times with PBS, and samples were mounted with glycerol/PBS (4:1). Then, cells were observed under a Leica TCS SP5 inverted confocal laser scanning microscope equipped with an argon–helium–neon laser light source (Leica Microsystems GmbH, Solms, Germany).

### 2.7. Flow Cytometric Analysis

SL-1 cells were transfected with pIE2-SlApaf-1-EGFP plasmid or siRNA targeting *Sl-apaf-1* for 48 h. Then, cells were treated with 500 ng/mL Act D for another 4 h. The cells were washed with PBS buffer twice, and 10^6^ cells were collected in 1.5 mL Eppendorf tubes by centrifugation at 500 *g* for 2 min. The cells were stained with Annexin V-APC/PI apoptosis detection kit (KeyGEN BioTECH, Nanjing, Jiangsu, China). Then, the stained cells were analyzed on a BD Coulter Epics flow cytometer (Beckman Coulter Inc., San Diego, CA, USA).

### 2.8. Expression and Purification of Proteins in E. coli

The encoding sequences of the caspase recruitment domain of Sl-Apaf-1 (named Sl-Apaf-1-CARD, 1-88aa) and Sl-caspase5 (named Sl-caspase5-CARD, 1-132aa) were amplified with specified primers (Table 1) and cloned in the pET-22b vector with Nde I and Xho I. The CDS of *Sl-caspase-1* and *Sl-caspase-5* were also amplified with specific primers (Table 1) and cloned into the pET-22b vector. To prevent procaspase-1 self-cleavage at high concentrations, the C178 in Sl-caspase-1 was used to perform mutation, using the QuikChange II site-directed mutagenesis kit (Stratagene, La Jolla, CA, USA), and the expressed protein was named Sl-caspase-1 (C178A). All proteins were expressed with His tag.

The expression vector was transferred into *E. coli* BL21(DE3), and the strain was cultured in 3.2 L Luria-broth (LB) media with 100 μg/mL ampicillin at 37 °C for 3 h until the OD_600_ value reached 1.0, then, induced by 0.4 mM isopropyl-1-thio-D-galactopyranoside (IPTG) at 28 °C for 24 h. To prevent procaspase-5 auto-activity, *E. coli* BL21 with caspase-5 expression vector was induced with IPTG at 28 °C for only 45 min. The cells were harvested, flash frozen in liquid nitrogen, and stored at −80 °C.

For the purification of the His tagged protein, *E. coli* cells were washed and re-suspended in 50 mL binding buffer (300 mM NaCl, 50 mM Tris, 20 mM imidazole, 2 mM PMSF, and 1 mM dithiothreitol, at pH 7.9). The bacterial cells were lysed with an ultra-high-pressure homogenizer at 1.2 × 10^9^ Pa three times at 4 °C. The cell lysate was centrifuged at 20,000× *g* for about 1 h to remove cellular debris. The fusion protein was purified by Ni-affinity resin (Merck-Novagen, Darmstadt, Germany) with elution buffer (300 mM NaCl, 50 mM Tris, 200 mM imidazole, 2 mM PMSF, and 1 mM dithiothreitol, at pH 7.9). The proteins were further purified on a superdex 16/60 200HR gel filtration column (GE Healthcare, Pittsburgh, PA, USA) with HEPES buffer (50 mM HEPES, 150 mM NaCl, DTT 2 mM, at pH 7.5). Purified proteins were then lyophilized and stored at −80 °C.

### 2.9. Protein Cleavage Experiments

The reaction was conducted at 28 °C for 30 min with 10 nmol dATP, 0.5 μg purified Sl-proaspase-5, 2 μg purified Af627, and 10 ug purified Sl-procaspase-1 (C178A) protein in the reaction buffer (100 mM KCl, 20 mM HEPES, 5 mM DTT, at pH 7.5) [12]. The reaction was stopped by adding 5× SDS-loading buffer and the mixture was boiled at 100 °C for 5 min. The reaction products were subjected to 12% SDS-PAGE. The results were visualized by Coomassie blue staining.

### 2.10. Isothermal Titration Calorimetry Assay

The concentration of the Sl-Apaf-1-CARD protein was adjusted to 50 µM and the Sl-caspase-5-CARD protein was adjusted to 450 µM with reaction buffer. The reaction buffer consisted of HEPES 10 mM, NaCl 100 mM, DTT 2 mM, at pH 7.4. The isothermal titration calorimetry assay (ITC) was performed with a VP-ITC isothermal titration calorimeter (GE Healthcare, Piscataway, NJ, USA). Parameters include: total injection 20, cell temperature 20 °C, reference power 5, initial delay 60 s, and stirring speed 500 RPM. The data were analyzed using ORIGIN 7.0 (OriginLab Corporation, Northampton, MA, USA).

### 2.11. Statistical Analysis

Statistical analyses were performed using GraphPad PRISM^®^ software version 6.0 (GraphPad Software Inc., San Diego, CA, USA). Data are expressed as the means ± SD. The results were considered statistically significant if *p* < 0.05 was obtained by the Student’s *t*-test.

### 2.12. GenBank Accession Numbers

*Sl-apaf-1*: MT793718; *Sl-caspase-1*: JN794535; *Sl-caspase-5*: JQ768051.

## 3. Results

### 3.1. Sequence and Phylogeny of Sl-Apaf-1

The complete cDNA sequence of *S. litura Apaf-1* was 5221 bp (termed as *Sl-apaf-1*) containing a 4626 bp open reading frame. The sequence encoded 1542 amino acids, a 0.2 kb untranslated region at the 5′ end, and a 0.4 kb untranslated region at the 3′ end (Figure 1). The deduced Sl-Apaf-1 has a computed molecular mass of 175.8 kD. Domain analysis with SMART indicated that the Sl-Apaf-1 protein contains the CARD domain at the C-terminus, followed by a NOD domain, and two WD domains at the N-terminus (Figure 2A). The predicted tertiary structure of Sl-Apaf-1 is shown in Figure 2A. The structure of the Sl-Apaf-1-CARD domain is displayed on the left (C-score = 0.4, TM-score = 0.77 ± 0.1) and the structure of Sl-Apaf-1 without CARD domain is displayed on the right (C-score = −0.81, TM-score = 0.61 ± 0.14).

To identify the evolutionary relationship between Apaf-1, Apaf-1 from mammals, lepidopteran, dipteran, and nematodes were downloaded and a phylogenetic tree was constructed. Surprisingly, Apaf-1 from lepidopteran and mammals were located on the same large branch of the phylogenetic tree, while the Apaf-1 from *D. melanogaster* and *C. elegans* (CED4) were located on another large branch (Figure 2B).

Sequence alignment of amino acids of Sl-Apaf-1 with other orthologous proteins showed that it has a high homology with proteins from noctuid insects and a moderately high homology with proteins from other lepidopteran families, and shows no homology with proteins from other orders (Appendix A). For example, the protein sequences of Sl-Apaf-1 shared 92% identity with *Spodoptera exigua* Apaf-1, 60% identity with *B. mori* Apaf-1, 58% identity with *P. xylostella* Apaf-1, and only 19% identity with *D. melanogaster* Dark.

### 3.2. Effect of Silencing Sl-apaf-1 Expression on Act-D-Induced Apoptosis

To determine the function of Sl-Apaf-1, three siRNA were designed and synthesized (Table 2). SL-1 cells were transfected with the siRNA and control siRNA. The expression levels of *Sl-apaf-1* decreased 26.7%, 44.3%, and 20.3% against the control after treatment for 48 h, and siRNA-2-treated cells had the lowest expression level (Figure 3A). Then, the cells were treated with Act D at a final concentration of 500 ng/mL for a further 6 h. DAPI staining showed that the numbers of nuclear body fragments (a phenotype of apoptosis) in the siRNA-2 treatment group and the untreated group (DMSO) were significantly lower than in other groups (Figure 3B). Next, the apoptosis percentage of SL-1 cell treated by Act D at 500 ng/mL for 4 h were measured. The apoptotic rate of the siRNA-2-treatment group was significantly lower than that of the untreated group (*p* = 0.011, Figure 3C). Furthermore, the activities of Sl-caspase-1 were also detected by rabbit anti-Sf-caspase-1 antiserum. Cleavage of Sl-caspase-1 protein in Act D + siRNA-2-treated cells was undetectable. However, in the Act D, Act D + siRNA1, and Act D + siRNA3 treatment groups, the Sl-caspase-1 P19 subunit and/or the P25 subunit could be detected (Figure 3D). These assays showed that siRNA-2 effectively interfered with *Sl-apaf-1* expression, and the decrease of *Sl-apaf-1* expression protected SL-1 cells from Act-D-induced apoptosis.

### 3.3. Sl-Apaf-1 Overexpression Promoted Apoptosis

To confirm the apoptosis activation ability of Sl-Apaf-1, Sl-Apaf-1-GFP was transiently expressed in SL-1 cells. Microscopic analysis showed that the Sl-Apaf-1-GFP fusion protein was expressed in the cytoplasm of SL-1 cells at a transfection efficiency of about 16%, and GFP protein was expressed in the whole cell at a transfection efficiency of about 30% (Figure 4A). Once the transfection efficiency was confirmed, cells were treated with Act D (at final concentrations of 500 ng/mL for 4 h) for FACS analysis. The rates of apoptotic cells in the Sl-Apaf-1-GFP expression group (42.16% ± 6.32%) were far higher than in the GFP expression group (19.41% ± 1.49%) when treated with Act D (Figure 4B,C). These results indicated that overexpression of Sl-Apaf-1 significantly promoted Act-D-induced apoptosis of SL-1 cells. Interestingly, the transient expression of Sl-Apaf-1 in SL-1 cells without stimuli did not have an apoptosis promoting effect (Appendix A).

### 3.4. Sl-Apaf-1 Activation of Caspase-5—Caspase-1 Pathway In Vitro

Apaf-1 is a caspase activator which activates the initiator caspase; the subsequently effector caspase is cleaved in mammalian cells. To detect whether Sl-Apaf-1 can activate the caspase initiator in lepidopteran insect cells and relay the caspase cascade, the Sl-Apaf-1 protein fragment (Af627) was expressed and purified. This fragment contains a CARD domain and a NOD domain (627 amino acids). Sl-procaspase-5 (a homologue of mammalian caspase-9), and Sl-procaspase-1 (a homologue of mammalian caspase-3) in *E. coli* and were used to investigate whether Sl-Apaf-1 could induce caspase activation. As shown in Figure 5A, lanes 2 and 3 indicated that Sl-procapase-5 could not directly cleave the Sl-procaspase-1 in the absence of Sl-Apaf-1. In contrast, Sl-Apaf-1 could not directly activate Sl-caspase-1 without Sl-capase-5 and dATP (lanes 4 and 5 in Figure 5A). Only in the presence of Sl-capase-5, could ATP and Af627 activate Sl-caspase-5, thus leading to the processing of Sl-caspase-1 and the generation of a large subunit (of approximately 19 kDa) and a small subunit (of approximately 12 kDa) of caspase-1 (lane 6). These data showed that Sl-Apaf-1 regulated the activation of Sl-caspase-5, and the active Sl-caspase-5 cleaved and activated the downstream Sl-caspase-1.

### 3.5. Interaction of Sl-Apaf-1 and Sl-Caspase-5

The effective binding of Apaf-1 and caspase-5 is essential for the activation of procaspase-5. Both can bind using the CARD domain in both proteins. The CARD domains of Sl-Apaf-1 (Sl-Apaf-1-CARD) and Sl-caspase-5 (Sl-caspase-5-CARD) were expressed and purified with the *E. coli* system. The binding interaction between Sl-Apaf-1-CARD and Sl-caspase-5-CARD was determined by ITC titration. The ITC data showed that Sl-Apaf-1-CARD and Sl-caspase-5-CARD had a binding affinity of 1.28 × 10^6^ ± 0.34 × 10^6^ M^–1^, a ∆H of −1.55 × 10^4^ ± 0.24 × 10^4^ kcal/mol, and a ∆S of −24.8 ± 8.6 cal/mol/deg, which were appropriate values. The stoichiometry of the two proteins was 1, which is consistent with the known binding model in mammals (Figure 5B). Clearly, these thermodynamic parameters reflected the ability of the specific binding between Sl-caspase5-CARD and Sl-Apaf-1 CARD, which supports the hypothesis that Sl-Apaf-1 was an activator of Sl-caspase-5.

### 3.6. Sl-Apaf-1 Promotes Apoptosis in Both Insect and Mammalian Cells

To determine the functional conservatism of Sl-Apaf-1, Sl-Apaf-1 was over-expressed in Lepidopteran cells (Sf9) and mammalian cells (U2OS). FACS assay of transfected cells was performed after treatment with 500 ng/mL Act D for 4 h. The results showed that overexpression of Sl-Apaf-1 significantly promoted apoptosis in both cells. For both, the apoptosis rate was promoted by over 20% (22.02% for Sf9 cells and 32.33% for U2OS cells) when compared with EGFP-overexpressing cells (Figure 6A,B).

## 4. Discussion

Apaf-1-mediated apoptosis signaling transduction has been well-characterized in mammalian cells [10]. Although the role of the Apaf-1 homologue, Dark, in apoptosis and development have been extensively studied in *Drosophila* [35,36,37,38], the function of Apaf-1 in the apoptosis of Lepidopteran insect cells remained unknown. The present study identified an *apaf-1* in *S. litura*, named *Sl-apaf-1*, which represents a new member of the Apaf-1 family. Apaf-1 from lepidopteran and mammals was located on the same large branch of the phylogenetic tree and sequence analysis showed that the Sl-Apaf-1 contains three domains: a CARD domain, as well as NOD and WD motifs. It is likely that Sl-Apaf-1 exerts a conservative function during apoptosis (Figure 2). To determine whether Sl-Apaf-1 is involved in the apoptosis of SL-1 cells, S*l-apaf-1* was silenced by RNAi, which decreased the apoptosis level induced by Act D, and Sl-caspase-1 cleavage was also inhibited in the cells (Figure 3). Moreover, Act D induced apoptosis in SL-1, Sf9, and U2OS cells, which could be further strengthened by over-expression of Sl-Apaf-1 (Figure 4). These results indicated that Sl-Apaf-1 is important for apoptosis in lepidopteran cells.

In the mitochondria-mediated apoptosis pathway, it is clear that Apaf-1 is an adaptor molecule that interacts with cytochrome c through its WD-40 repeats, while it recruits and activates procaspase-9 through its CARD. In *Drosophila*, studies have demonstrated that Dark plays a central role during apoptosis and appears to act independent of cytochrome c [35,38]. In this study, Apaf-1 from both lepidopteran and mammal cells was located on the same large branch of the phylogenetic tree, while Dark was located on another branch (Figure 2). Sequence analysis showed that Sl-Apaf-1 has a higher sequence similarity with mammalian orthologous protein than Dark, and Sl-Apaf-1 shared no homology with Dark. Based on our previous report on cytochrome c [26], the release of cytochrome c is an important step during the apoptosis of *S. litura* cells. Furthermore, Sl-Apaf-1 could not promote apoptosis if it was overexpressed in Sf9 cells without stimuli. It is likely that Sl-Apaf-1 functions in a similar way with mammalian Apaf-1 during apoptosis, and cytochrome c is an important factor for the activation of Sl-Apaf-1. The interactions between Apaf-1 and cytochrome c were tested, and they could co-localize in human U2OS cells by co-transfection with plasmids expressing Sl-Apaf-1 or *S. litura* cytochrome c with fluorescent biological labels (Appendix A). However, because of the low level of Apaf-1 in SL-1 cells and the difficulty to express and completely purify Apaf-1, the evidence to demonstrate that cytochrome c activates mitochondrial apoptosis by binding Apaf-1 was insufficient. More studies are needed to clarify this point.

As an adaptor or activator, in mammals, Apaf-1 stimulates caspase-9 activation through the CARD domain and formed a 1:1 complex between the CARD domains of Apaf-1 and caspase 9 [7,39]. Interaction between Apaf-1 CARD and caspase-9 CARD is indispensable for caspase-9 activation, which, in turn, promotes caspase-3 activity. In this study, ITC assay showed that the binding affinity for the Sl-Apaf-1-CARD/Sl-caspase-5-CARD was relatively high and had a stoichiometry of 1:1. This indicated that Sl-Apaf-1 and Sl-caspase-5 have binding potential under certain conditions (i.e., in the presence of cytochrome c and dATP, Figure 5). It has been reported that activated caspase-5 (Dronc) could directly activate caspase-1 and promote apoptosis in *S. frugiperda*, *B. mori*, and *Lymantria dispar* [29,40,41]. Furthermore, the present study showed that auto-activated Sl-caspase-5 could cleave the Sl-caspase-1 in another assay (Appendix A). Interestingly, Af627 (i.e., Sl-Apaf-1 without the WD40 domains) could active Sl-caspase-5 in the presence of ATP, and this activated Sl-caspase-5 could, in turn, directly activate the Sl-procaspase-1 in the in vitro assay. These results strongly indicated that Sl-Apaf-1 follows the classic model of the mitochondrial apoptosis pathway in mammals.

## 5. Conclusions

An *apaf-1* homolog gene from *S. litura* was identified and designated as *Sl-apaf-1*. Sl-Apaf-1 can bind and activate Sl-caspase-5. These results indicate that Sl-Apaf-1 plays a key role in apoptosis, suggesting that the apoptosis pathway is conserved between Lepidopteran insects and mammals.

## Figures and Tables

**Figure 1 insects-12-00064-f001:**
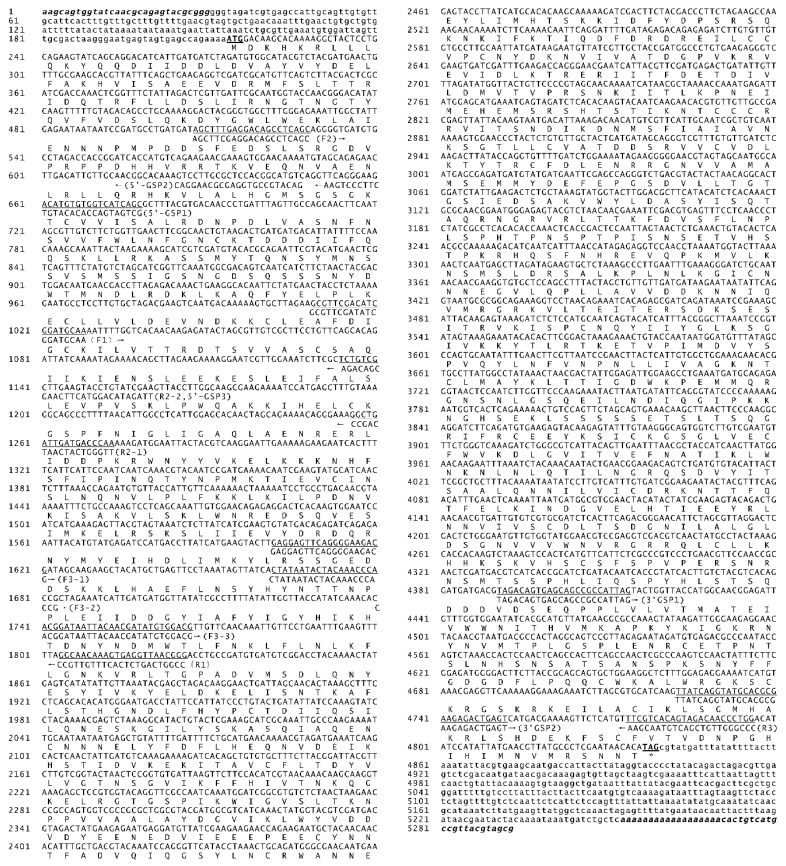
Sequence of S*l-apaf-1* cDNA and predicted amino acids. The complete cDNA consists of 5221 bp and contains an open reading frame of 4626 bp that encodes 1542 amino acids. The start codon and stop codons are underlined.

**Figure 2 insects-12-00064-f002:**
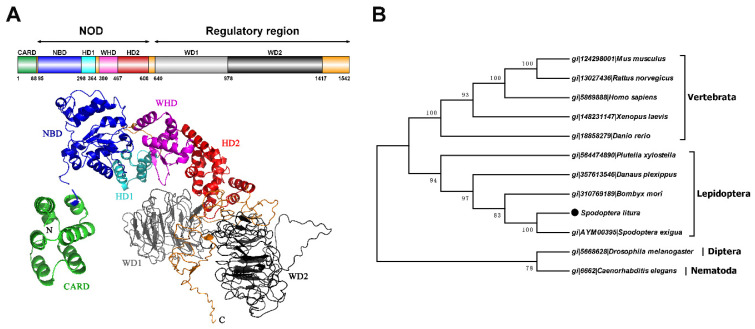
Predicted tertiary and domain structures of Sl-Apaf-1 protein and corresponding phylogenetic tree. (**A**) is a schematic representation and prediction of the tertiary structure of the Sl-Apaf-1 protein. Subdomains are color coded, and linkers are shown in orange. I-Tasser was used to predict the tertiary structures of the putative Apaf-1 protein, the first of model was the most accurate and was therefore used for analysis. (**B**) Evolutionary tree of the Apaf-1 protein based on the neighbor-joining method with MEGA 7.0.21 software.

**Figure 3 insects-12-00064-f003:**
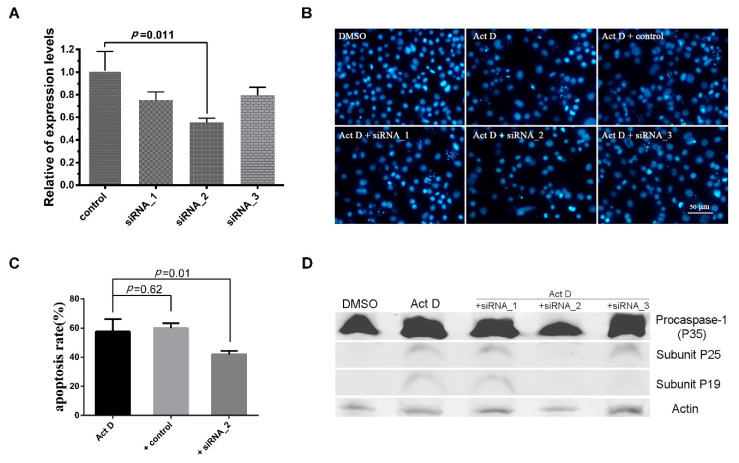
RNAi analysis of the apoptosis activator of Sl-Apaf-1. (**A**) Analysis of the *Sl-apaf-1* expression levels by qRT-PCR; (**B**) Microscopic analysis of SL-1 cells; (**C**) FACS analysis of SL-1 cells treated with Act D; (**D**) Western blot analysis of Sl-caspase-1 activation. Original Western blot see Appendix A.

**Figure 4 insects-12-00064-f004:**
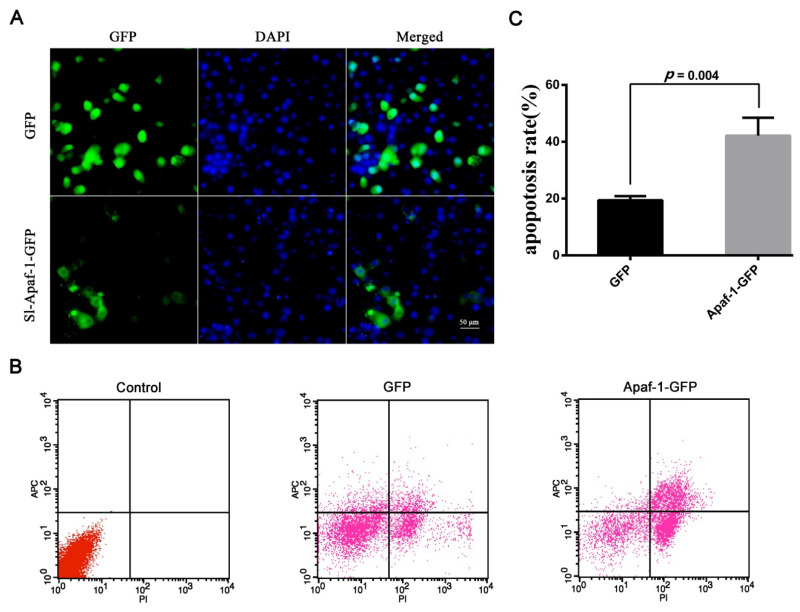
Sl-Apaf-1 overexpression in the SL-1 cell line. (**A**) Microscopic analysis of Apaf-1-GFP expression in SL-1 cells; (**B**) FACS analysis of SL-1 cells transfected with Apaf-1 overexpression vector, Left: Normal SL-1 cells; Middle: SL-1 cells transfected with GFP expression vector; Right: SL-1 cells transfected with the Sl-Apaf-1-GFP fusion expression vector; (**C**) Act-D-induced apoptosis of SL-1 cells when Sl-Apaf-1 was overexpressed.

**Figure 5 insects-12-00064-f005:**
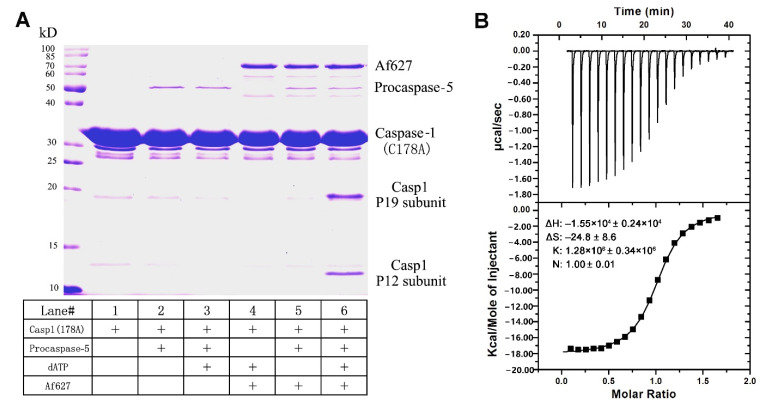
Sl-Apaf-1 triggers Sl-caspase-5 activation in vitro according to the ITC assay. (**A**) Sl-Apaf-1, Sl-caspase-5, and Sl-caspase-1 (Casp1) were expressed and purified. These proteins were incubated under different treatments and then separated by SDS-PAGE analysis. Activity was assayed by cleavage of Sl-caspase-1. Sl-caspase-1 is shown in lane 1. Sl-procaspase-5 could not directly activate the Sl-caspase-1 (lines 2 and 3). Sl-Apaf-1 could not directly trigger Sl-caspase-1 activation in the absence of ATP and procaspase-5 (lanes 4 and 5). Only in the presence of both procaspase-5 and dATP, could Sl-Apaf-1 activate Sl-caspase-1, which lead to the processing of Sl-caspase-1 to generate the p19 large subunit and the p12 small subunit (see lane 6). (**B**) ITC assay of the interaction between the Sl-Apaf-1-CARD domain protein fragment and the Sl-caspase-5-CARD domain protein fragment. Upper panel, interaction of the Sl-Apaf-1-CARD domain with the Sl-caspase-5-CARD domain. Lower panel, control: Reaction buffer interaction with caspase-5-CARD domain. Upper panel—raw ITC data, lower panel—integrated ITC data with the curve.

**Figure 6 insects-12-00064-f006:**
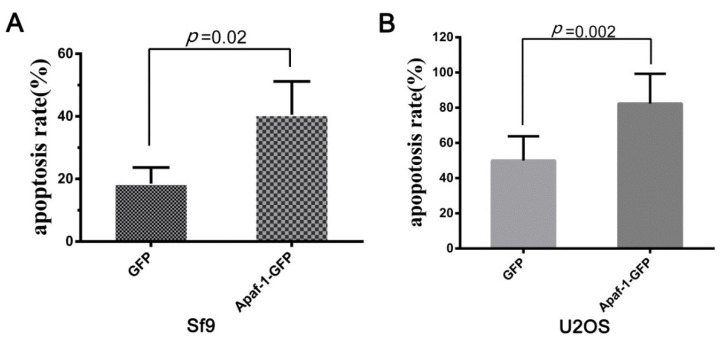
Act-D-induced apoptosis of insect cells (Sf9, (**A**)) and mammalian cells (U2OS, (**B**)) when Sl-Apaf-1 was overexpressed.

**Table 1 insects-12-00064-t001:** Primers used in this study.

Application	Gene	Sequences (5′–3′)
qPCR		
	*Sl-apaf-1*	ACGATGAACTGTTTGCGAAGC
		ATTTCTCCCAAAGCCACCC
	*Sl-GADPH*	GCCACCACCGCTACCCAGAA
		GGAACACGGAAAGCCATAC
Protein expression		
	*Sl-Af88*	CCGCATATGGACAAGCACAAAAGGCTACTC
		TAGCTCGAGAGCCAATTTCTCCCAAAGCCAC
	*Sl-Af627*	CCGCATATGGACAAGCACAAAAGGCTACTC
		CTGGCGGCCGCTTCATCTACGTTTTGTTCATGC
	*Sl-apaf-1*	AATCTCGAGCCACCATGGACAAGCACAAAAGGCTAC
		AATGGATCCGTGTTATTCGAGCGCATAACGTTC
	*Caspase-1*	AGCCATATGGCGGACGGAAAACAAGAC
		TGTCTCGAGCTTTTTACCAAACACAAG
	*Caspase-5*	GAACCATATGCAAAGAGAACACAGGGATG
		CAGGAATTCACTCGTAGAGACCGGGG
	*Caspase-5 (CARD)*	GAACCATATGCAAAGAGAACACAGGGATG
		TCACTCGAGAACGGCAGGAGGAGGTGGTGTAG

**Table 2 insects-12-00064-t002:** SiRNAs used in this study.

siRNA	Sequences (5′–3′)
Control	UUCUCCGAACGUGUCACGUTT
	ACGUGACACGUUCGGAGAATT
siRNA_1	GGAUGUUAUCGAAGAAGAATT
	UUCUUCUUCGAUAACAUCCTT
siRNA_2	GCUUCAUACAUCUCACAAATT
	UUUGUGAGAUGUAUGAAGCTT
siRNA_3	GGGCGUCAUUACAGUUGAATT
	UUCAACUGUAAUGACGCCCTT

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
