# Peer review of "Identification and Functional Analysis of Apoptotic Protease Activating Factor-1 (Apaf-1) from Spodoptera litura"

_insects, 2021, doi:10.3390/insects12010064_

Round 1

Reviewer 1 Report

Apoptotic signal pathways are well investigated in mammalian cell and Drosophila melanogaster  cell, and there are differences between them. Cytochrome C might not be involved in apoptosis of Drosophila melanogaster cells. The previous reports revealed that apoptosis of lepidopteran cells is similar with that of mammalian cells. For example, cytochrome c is involved in apoptosis of lepidopteran cells. Although several caspases have been characterized in lepidopteran insects. However, Apaf-1 is not well documented in lepidopteran cells. The manuscript reported the characterization and function of Apaf-1 from Spodoptera litura at many aspects. Especially, the results clearly showed that binding affinity of Apaf-1 to homologue of mammalian caspase 9 by ITC and its involvement in apoptosis activation using WB. The results are valuable and enrich the theory of apoptosis of lepidopteran insect cells.

However, I also have suggestion for improvement of the manuscript.

  1. Check carefully the spelling and grammar.
  2. Which tissues are the two used cell lines established from ?
  3. Figure 1, I did not see the underlined starting code and stop code.
  4. Line “replaced the” should be “the replaced”。
  5. Line 195, “expression and purify” should be “expressed and purified”.
  6. Line 305, “expression” should be “expressed”.
  7. Line 346, remove “liu et al.”.
  8. Line 353, “purified” should be “purification”.
  9. Line 353, “Completly” should be “complete”.
  10. Line 370, “Spodopteran” should be “S.”

Author Response

Response : Thank you very much. We have corrected the manuscript as required.

  1. Check carefully the spelling and grammar.

   Response: Yes. We have asked a professional to correct it.

  1. Which tissues are the two used cell lines established from ?

Response: We have added them in the rewritten manuscript.

  1. Figure 1, I did not see the underlined starting code and stop code.

   Response: We have added them in Figure 1.

  1. Line “replaced the” should be “the replaced”。

Response: We have changed it.

  1. Line 195, “expression and purify” should be “expressed and purified”.

       Response: We have corrected them.

  1. Line 305, “expression” should be “expressed”.

       Response: We have changed it.

  1. Line 346, remove “liu et al.”.

     Response: We have removed it .

  1. Line 353, “purified” should be “purification”.

      Response: We have corrected it.

  1. Line 353, “Completly” should be “complete”.

    Response: We have corrected it.

  1. Line 370, “Spodopteran” should be “S.”

    Response: We have changed it.

Reviewer 2 Report

Comments for insects-1020265

In the manuscript, “Identification and function analysis of apoptotic proteases activating factor-1 (Apaf-1) from Spodoptera liutra”, Ma and co-workers proposed that Apaf-1 is an adopter protein in the apoptotic signaling pathway initiated apoptosis with downstream molecular caspases. They found that Apaf-1 is evolutionally conserved in the apoptosis signaling pathway. Overexpression of Apaf-1 enhanced apoptosis induced by ActD in two insect cell lines and a mammalian cell line; on the contrary, siRNA silenced Apaf-1 decreased apoptosis. In general, the paper is interesting but the writing of this manuscript is not clear. And some data loss control and poor quality figures. Thus, I believe this manuscript is not ready to publish. Some suggestions are listed below for the authors’ consideration.

  1. The authors should combine Fig. 1 and 2 to make the paper more logical such that it shows the importance of the domains and presented sequences.
  2. It would be better to change the table 1 primer part in the text in methods and insert the table 1 siRNA part into the 3.2 section.
  3. It would be helpful to add control in Figure 3. A and redo C; also Figure 4. A look likes not in the same area, etc.
  4. The authors should read this manuscript carefully to correct some wrong descriptions.

Author Response

It would be better to change the table 1 primer part in the text in methods and insert the table 1 siRNA part into the 3.2 section.

Response: Thanks,we have done it in the re-submitted manuscript.

It would be helpful to add control in Figure 3. A and redo C; also Figure 4. A look likes not in the same area, etc.

Response: Thanks, we have improved the Figure 3A , 4A, and redone C.

The authors should read this manuscript carefully to correct some wrong descriptions.

Response: Thanks, we have checked the manuscript, and the new manuscript has been revised by professionals.

Reviewer 3 Report

This is an interesting study. The authors used a series of well-designed experiments to investigate the function of Apaf-1 in a Lepidopteran insect, and demonstrate its role in response to mitochondrial apoptosis signal and activation of apoptosis. Sl-Apaf-1 mainly binds the initiator Sl-caspase-5, and then activates the signaling of effector caspase. This signaling pathway is conservative in lepidopteran insects and mammals (cells). The experiments were well rounded to provided evidence of the proposed idea. The methods section was well written and easy to follow.

The authors already stained cells with Annexin/PI; it would be better if the authors add TUNEL assay to show the apoptotic cells. For instance, for Fig. 3B. But it’s fine if this cannot be done in a short time for this manuscript.

Minor comments:

L39-43: It would be better if the authors add information in mammals like: The extrinsic pathway is mediated by caspase 8 while the intrinsic pathway can be initiated through caspase 9. In addition, these pathways trigger apoptosis through the cleavage of the downstream effector proteins, caspase 3 and 7. Or something like that.

L43: in the text, it is still necessary to give the full name of Apaf-1 for the first time.

L64-66: This sentence is redundancy. Please rephrase it.

L127-132: Please provide references for software like MEGA.

L165: Although this showed in the abstract, please also give the reader what is Act D and it is an inducer or inhibitor of apoptosis.

L226: Please italic D. melanogaster and C.elegans. Also, check throughout the manuscript. e.g., L240, L327, L341…

L240: This is interesting. Only 11% of identity with Drosophila Apaf-1. This is Dark?

L250-L253: Please reverse Fig. 3D and Fig. 3C.

L302: Correct Sl-caspas-5 >> Sl-caspase-5 Please check the spell of caspase throughout the manuscript. I found several places wrong. E.g., L359, L278, L285…

Author Response

The authors already stained cells with Annexin/PI; it would be better if the authors add TUNEL assay to show the apoptotic cells. For instance, for Fig. 3B. But it’s fine if this cannot be done in a short time for this manuscript.

Response: Thanks. We renewed the explanation to make the expression more rigorous.

Minor comments:

L39-43: It would be better if the authors add information in mammals like: The extrinsic pathway is mediated by caspase 8 while the intrinsic pathway can be initiated through caspase 9. In addition, these pathways trigger apoptosis through the cleavage of the downstream effector proteins, caspase 3 and 7. Or something like that.

Response: Thanks. We have rewritten it as shown in text.

L43: in the text, it is still necessary to give the full name of Apaf-1 for the first time.

Response: Thanks. We have revised it.

L64-66: This sentence is redundancy. Please rephrase it.

Response: Thanks. We have deleted it.

L127-132: Please provide references for software like MEGA.

Response: Thanks. We have provided it.

L165: Although this showed in the abstract, please also give the reader what is Act D and it is an inducer or inhibitor of apoptosis.

Response: Thanks. We have provided the mechanism of Actinomycin D (ActD) inducing the apoptosis.

L226: Please italic D. melanogaster and C.elegans. Also, check throughout the manuscript. e.g., L240, L327, L341…

Response:  Thanks. We have revised it throughout the manuscript.

L240: This is interesting. Only 11% of identity with Drosophila Apaf-1. This is Dark?

Response:  Yes. It is Dark. We have revised it.

L250-L253: Please reverse Fig. 3D and Fig. 3C.

 Response: Thanks. We have reversed it in new manuscript.

L302: Correct Sl-caspas-5 >> Sl-caspase-5 Please check the spell of caspase throughout the manuscript. I found several places wrong. E.g., L359, L278, L285…

Response: Thanks. We have revised it.

Reviewer 4 Report

The main aim of this research article by Ma and colleagues, was to identify and characterize the Apaf-1 protein in Spodoptera litura and to show that Sl-Apaf-1 interacts and activates Sl-caspase-5, which (Sl-caspase-5) in turn, activates Sl-caspase-1. Although, the findings should be of interest to researchers working with apoptotic mechanisms in Lepidopteran insects, the whole paper is seriously downgraded by the quality of its presentation. I believe that if the article is thoroughly corrected and edited (in terms of grammar, syntax, punctuation, and spelling) by a native speaker of the English language, who also happens to be an expert in this scientific field, it is a promising work. Unfortunately, in its present form is not suitable for publication.

Except from the English language, which, in some sentences is so poor that the sentence is very difficult to understand (i.e. lines 30, 121, 141, 176, 280, 295, 304, 305, 330, 333, 336, 353, 355), other issues that must be addressed are:

In the Materials and Methods section, the following issues should be addressed:

  • The catalogue numbers of Cell Signaling Technology antibodies should be provided.
  • It is I-Tasser and not 1-Tasser. References and web addresses for all the bioinformatic tools used must be provided.
  • A section for microscopy is needed.
  • Line 93: #28 does not refer to Paul Friesen.
  • Line 128: please provide ClustalW alignment as a Supplementary figure.
  • Line 175: #31 is not relevant.
  • Lines 201 and 202: why these concentrations were used?
  • Line 212: MT793718 cannot be found in GenBank

In the Results section:

  • Figure 1: An image with better resolution must be provided, and the primer pairs must be somehow highlighted for better comprehension.
  • Figure 2B: In order to estimate the quality of the predicted models by I-TASSER (not 1-Tasser), C-score and TM-score must be provided.
  • A short sentence explaining the mechanism of Actinomycin D (ActD) action is required.
  • Figure 3B: What are the differences, please explain.
  • DAPI staining is not a valid apoptotic marker. It only shows chromatin condensation. It should be used in combination with a more appropriate apoptotic marker (i.e. TUNEL/acridine orange, DNA ladder formation, caspase activation)
  • Figure 3C: Western blot analysis needs normalization.
  • Line 264: Please explain the differences in the time of transfection.
  • Line 271: Please provide the relevant data as a supplementary figure.
  • Figure 4A is not well explained.
  • Figure 5A: Please discuss the Sl-Casp1 activation by Sl-Apaf-1 without the participation of Cyto-c and also why in lane 6 there is so much uncleaved Sl-Casp1.
  • Figure 5B: Please provide an image with better resolution and discuss the results.
  • Figure 6B: what does VEL stand for?
  • Please explain the rationale of using U2OS cell line (and not another mammalian cell line) and analyze the experiment.
  • Spodoptera litura Apaf-1, is other times reported as Sl-Apaf-1 and other times as Spli-Apaf-1 or SlApaf-1, please choose (preferably Sl-Apaf-1)

In the Discussion section:

  • Line 328: reference #32 is not relevant.
  • Lines 352 and 365: Please provide the data as supplementary figures.

In the References section:

  • Line 75: the correct reference is #30 and not #29.
  • Refs #1 and 20 are not complete.
  • Please correct Refs #32, 33, 37, 38 and 39.

Author Response

Except from the English language, which, in some sentences is so poor that the sentence is very difficult to understand (i.e. lines 30, 121, 141, 176, 280, 295, 304, 305, 330, 333, 336, 353, 355)

Response: Thanks. The new manuscript has been revised by a professionals.

In the Materials and Methods section, the following issues should be addressed: The catalogue numbers of Cell Signaling Technology antibodies should be provided.

Response: Thanks. We have provided the catalogue numbers of CST antibodies in the corrected manuscript.

It is I-Tasser and not 1-Tasser. References and web addresses for all the bioinformatic tools used must be provided.

Response: Thanks. We have corrected it and provided the reference and web addresses.

A section for microscopy is needed.

Response: Thanks. We have provided the section in Materials and methods.

Line 93: #28 does not refer to Paul Friesen.

Response: Thanks. We have corrected it.

Line 128: please provide Clustal W alignment as a Supplementary figure.

Response: Thanks. We have provided Clustal W alignment in supplementary file.

Line 175: #31 is not relevant.

Response: Thanks. We have corrected it.

Lines 201 and 202: why these concentrations were used?

Response: Thanks. We have provided the references.

Line 212: MT793718 cannot be found in GenBank

 Response: Thanks.We have written letter to GenBank to release the data.

In the Results section:

Figure 1: An image with better resolution must be provided, and the primer pairs must be somehow highlighted for better comprehension.

Response: Thanks. We have improved it.

Figure 2B: In order to estimate the quality of the predicted models by I-TASSER (not 1-Tasser), C-score and TM-score must be provided.

Response: Thanks. We have provided it.

A short sentence explaining the mechanism of Actinomycin D (ActD) action is required.

Response: Thanks. We have provided it.

Figure 3B: What are the differences, please explain.

Response: Thank you. We have improved the expression in the new manuscript.

DAPI staining is not a valid apoptotic marker. It only shows chromatin condensation. It should be used in combination with a more appropriate apoptotic marker (i.e. TUNEL/acridine orange, DNA ladder formation, caspase activation)

Response: Thanks. We have revised the expression in the new manuscript to make the description more rigorous.

Figure 3C: Western blot analysis needs normalization.

Response: Thanks. The specificity of the antiserum used in this study is not very high, and data normalization may produce inaccurate results.

Line 264: Please explain the differences in the time of transfection.

Response: Thanks. There were some clerical errors, and we have corrected it in new manuscript.

Line 271: Please provide the relevant data as a supplementary figure.

Response: Thanks. We have provided the relevant data in the supplementary file.

Figure 4A is not well explained.

Response:Thanks. We re-described Figure 4A in the new manuscript.

Figure 5A: Please discuss the Sl-Casp1 activation by Sl-Apaf-1 without the participation of Cyto-c and also why in lane 6 there is so much uncleaved Sl-Casp1.

Response: Thanks. Casp1 activation by Apaf-1 without the participation of Cyto-c had been reported (#13), Figure 5A is just to provide evidence that Sl-Apaf-1 and mammalian Apaf-1 have the same biochemical function.

Sl-caspase-1 is cleaved and activated by Sl-caspase-5, and only a little Sl-procaspase-5 was used in the reaction system to avoid the self-activation of Sl-procaspase-5, therefore, no sufficient Sl-caspase-1 was cleaved.  

Figure 5B: Please provide an image with better resolution and discuss the results.

Response: Thanks. We have provided a better image in the new manuscript, and discussed it.

Figure 6B: what does VEL stand for?

Response: Thanks. VEL is EGFP tag with an amino acid mutation, we have revised the expression to make the meaning clearer in the new manuscript.

Please explain the rationale of using U2OS cell line (and not another mammalian cell line) and analyze the experiment.

Response: U2OS cell line is suitable for ultra-high resolution imaging (Fig. S3). Here, it is just a representative of mammalian cells.

Spodoptera litura Apaf-1, is other times reported as Sl-Apaf-1 and other times as Spli-Apaf-1 or SlApaf-1, please choose (preferably Sl-Apaf-1)

Response:  Thanks. We used the “Sl-Apaf-1” throughout the new manuscript.

In the Discussion section:

Line 328: reference #32 is not relevant.

Response: Thanks. We have corrected the reference.

Lines 352 and 365: Please provide the data as supplementary figures.

Response:  Thanks. We have provided the data of Line 352 in supplementary file. The data of line 365 were provide as follows.

In the References section:

Line 75: the correct reference is #30 and not #29.

Response: Thanks. We have corrected the reference.

Refs #1 and 20 are not complete.

Response: Thanks. We have completed the reference.

Please correct Refs #32, 33, 37, 38 and 39.

Response: Thanks. We have corrected the reference.

Round 2

Reviewer 2 Report

The authors have addressed my concerns.

Author Response

No suggestion.

Reviewer 4 Report

The manuscript in its present form, is significantly improved

Minor comments:

Please correct the sentences in the following lines, preferably:

line 45: The extrinsic pathway is mediated by death receptors.

line 72: the cleavage of P25 at D28

lines 298-300: This fragment contains a CARD domain and a NOD domain (627 amino acids). Sl-procaspase-5 (a homologue of mammalian caspase-9), and Sl-procaspase-1 (a homologue of mammalian caspase-3) in E. coli were used to investigate whether Sl-Apaf-1 could induce caspase activation.

line 312: were expressed

line 348: represents a new member

line 367: similar way with mammalian

Author Response

Dear Editor Hang and the reviewers,

Thank you for the careful check and providing the correction. We have put the correction into the modified version R2, which were labeled by Green.

Best Regards,

Kaiyu Liu

Minor comments:

Please correct the sentences in the following lines, preferably:

line 45: The extrinsic pathway is mediated by death receptors.

Response: Put it into the line 45.

line 72: the cleavage of P25 at D28

Response: Put it into the line 72.

lines 298-300: This fragment contains a CARD domain and a NOD domain (627 amino acids). Sl-procaspase-5 (a homologue of mammalian caspase-9), and Sl-procaspase-1 (a homologue of mammalian caspase-3) in E. coli were used to investigate whether Sl-Apaf-1 could induce caspase activation.

Response: Put it into the line 4298-300.

line 312: were expressed

Response: Put it into the line 312.

line 348: represents a new member.

Response: Put it into the line 348.

line 367: similar way with mammalian.

Response: Put it into the line 367.